# A Geometric Analysis of Logit Embeddings for Out-of-Distribution Detection

## Abstract

Out-of-distribution (OOD) data pose a significant challenge to deep learning (DL) classifiers, prompting extensive research into their effective detection methods. Current state-of-the-art OOD detection methods usually employ a scoring technique designed to assign lower scores to OOD samples compared to in-distribution (ID) ones. Nevertheless, these approaches lack foresight into the configuration of OOD and ID data within the latent space. Instead, they make an implicit assumption about their inherent separation or force a separation post-training by utilizing selected OOD data. As a result, most OOD detection methods result in complicated and hard-to-validate scoring techniques. This study conducts a thorough analysis of the logit embedding landscape, revealing that the ID and OOD data exhibit a distinct spatial configuration. Specifically, we empirically observe that the OOD data are drawn to the center of the logit space. In contrast, ID data are repelled from the center, dispersing outward into distinct, class-wise clusters aligned along the orthogonal axes that span the logit space. This study highlights the critical role of the DL vision-based classifier in differentiating between ID and OOD logits.

## 1 Introduction

Deep learning (DL) (vision-based) classifiers perform well at generalizing from large datasets, achieving superior classification accuracy compared to many alternatives. They deliver highly accurate predictions when the test data aligns with the training data distribution. However, current DL classifiers are not capable of handling out-of-distribution (OOD) data. This limits their application in critical fields such as autonomous systems. For instance, in autonomous driving, vision-based DL classifiers are used to identify traffic signs, vehicles, and pedestrians from camera feeds. At test time the car may encounter previously unseen objects, e.g., fallen cargo or unusual construction equipment, which constitute OOD inputs. If the perception module wrongly assigns these novel obstacles to familiar classes, the planning stack can issue unsafe actions, jeopardising road safety. Recent OOD detection methods predominantly operate under the assumption that a classifier, when trained on in-distribution (ID) data, intrinsically maps the logits of OOD samples to a distinct spatial location within the logit landscape, divergent from those of ID instances. Thus, differentiating OOD instances from ID data typically involves assigning high likelihood values to the logit (or embeddings) location of the ID samples (Vyas et al., 2018; Lee et al., 2018; Sun et al., 2022; Gomes et al., 2022; Liu et al., 2020). Nevertheless, these strategies lack foundational awareness regarding the specific locational distribution of OOD samples in the embedding space. Consequently, these techniques attempt complicated and computationally intensive density estimations of the ID logits, categorizing those samples that fall beneath a certain likelihood threshold as OOD.

Instead, our study demonstrates that a well-trained DL classifier, incorporating non-linearities that suppress negative values (e.g., ReLU), systematically maps ID data into well-defined, class-specific clusters with a consistent spatial configuration. These ID clusters are situated along orthogonal axes within the positively constrained logit space and are notably separated from the logit space's center. Additionally, we empirically observe that OOD data are not arbitrarily scattered in the logit space but always drift toward the center. While prior work has leveraged ID logit to detect OOD data (Lee et al., 2018; Liu et al., 2020; Choi et al., 2024; Katz-Samuels et al., 2025), these approaches neither identify nor exploit the expected positioning of ID and OOD data within the logit space. Instead, this

work showcases empirically where ID and OOD logits are structurally positioned within the logit space.

The noted positioning of OOD and ID logits lays the groundwork for the possible creation of a binary classifier (OOD from ID), which could lead to simpler yet more effective OOD detection models. Key contributions of this study:

1. We showcase empirically that ID class clusters align along orthogonal axes in the positive logit space, shifted away from the center, while OOD data cluster near the origin.

2. We provide extensive empirical validation across multiple models, supported by an ablation study.

## 2 RELATED WORK

Although the detection of OODs through ID logits has been extensively studied (Wang et al., 2022; Liu et al., 2020; Hsu et al., 2020; Lee et al., 2018), existing methods do not consider their spatial configuration. Conventional OOD detection methods predominantly classify data by first identifying ID samples and subsequently labeling all other samples as OOD by default. A recent empirical investigation has not only highlighted the transferability of ID training strategies to OOD detection but also identified a tangible correlation between the robustness of ID training protocols and OOD detection efficacy (Wenzel et al., 2022). This study suggests that refining ID training methods could unlock potential pathways for enhancing OOD detection. Another study examines the influence of pre-trained ViT (Vaswani et al., 2017) on ImageNet and reports notable improvements in OOD detection performance (Dosovitskiy et al., 2021). Parallel to these observations, another line of research incorporates outlier data, surrogates for OOD samples, within the training phase. This is achieved through an auxiliary loss term that sharpens the contrast between ID and outlier inputs, potentially strengthening OOD detection (Katz-Samuels et al., 2025; Hendrycks et al., 2019; Wang et al., 2023; Du et al., 2022; Ming et al., 2022). Complementing these approaches, there has been a significant effort to restrict the classification of ID data into a hyperspherical embedding, which intrinsically helps OOD detection (Ming et al., 2023). Another line of research assumes an inherent separation between OOD and ID logits and tries to devise scoring techniques using solely ID logits or softmax output. The OOD detection works by classifying as OOD anything that is not ID. The earliest work on this front assumes clustering of ID logits into a multimodal Gaussian distribution and then tries to utilize Mahalanobis distance (Lee et al., 2018). More advanced methods try to upgrade the Mahalonobis distance with geometric information using the Fisher Information matrix (Gomes et al., 2022) Other works try to perform a data-driven density estimation using energy-based models (Liu et al., 2020). Another promising research study demonstrates the utility of enhanced Hopfield networks in amplifying the distinction between ID and OOD data (Doe et al., 2025). Similarly, another proactive work tries to increase the separability between OOD and ID using kernel principal component analysis on the OOD and ID embeddings (Fang et al., 2025). Last but not least, (Zhang et al., 2024) tries to learn the shape of the ID feature space using an online expectation maximization, which enhances the detection of OOD post-training.

## 3 METHOD

**Underlying assumptions:** In exploring ID and OOD data, it is crucial to delineate their distinctions in relation to DL classifiers. The training dataset is regarded as the optimal empirical representation of ID data. Under the manifold assumption (Bengio et al., 2013), ID data tend to aggregate into class-specific regions based on discriminative features corresponding to each class. While the exact parameterization of this feature space remains unknown, the annotated empirical ID dataset serves as a practical surrogate. For ID data classification, DL classifiers are optimized to generalize the distribution of class-specific discriminative features to effectively map the ID data into respective clusters within the logit vector space. Whenever we encounter data whose features reside outside the distribution of these class-specific discriminative features, they are considered OOD. The divergence of OOD features can vary, ranging from near-OOD (slightly shifted from ID data) to far-OOD (highly dissimilar). More concretely, a DL classifier trained to differentiate cats from dogs will encounter difficulties with photos of horses and wolves. However, because wolves' features are more similar to dogs, wolves represent near-OOD data compared to horses, which are farther from both trained categories. Consequently, the more distant the OOD features are, the less they resemble the discriminative features of ID classes, making them less perceptible by the DL classifier.

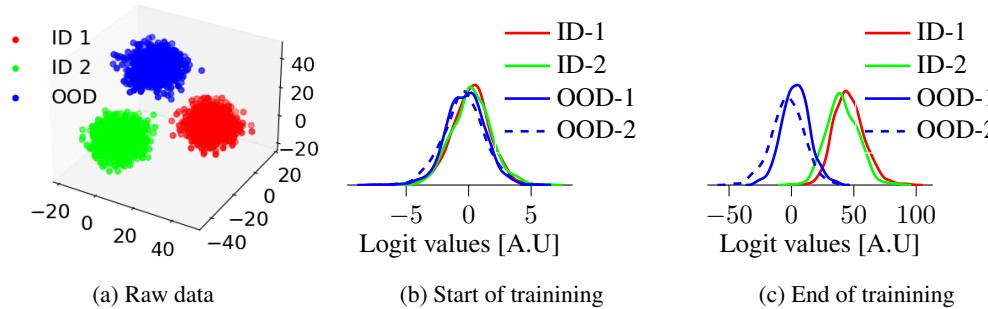

(a) Raw data      (b) Start of trainining      (c) End of trainining

Figure 1: Figure 1a shows raw training data sampled from a multimodal Gaussian distribution, utilized to train a simple MLP binary classifier depicted in Appendix B. In this figure, red and green points denote ID classes for binary classification, and blue points represent OOD data. Figure 1b and fig. 1c show kernel density estimations (KDE) across logit cells for both OOD and ID data before and after model training, respectively. In both figures, 'OOD-1' and 'OOD-2' refer to KDEs for OOD data within the first and second logits, while 'ID-1' and 'ID-2' represent KDEs for ID class one data in the first logit cell and ID class two data in the second logit cell, respectively.

**Allocation of OOD and ID logits pre-train:** To visually represent the empirical distributions of ID and OOD logits before and after the training phase, we employed a binary classification model using a multilayer perceptron (MLP) model (see table 1 in Appendix B). To minimize any initial biases, weights of DL classifiers are initialized using a centered Gaussian distribution (Glorot & Bengio, 2010), while the biases are set to zero. Hence, one can consider the features of both ID and OOD data distributionally shifted from the initialized weights (see Assumption 1). This shift manifests as reduced co-variation between the model's weight parameters and both OOD/ID data samples, resulting in a negligible expected dot product (see Corollary 1). Because DL classifiers fundamentally rely on the dot product operations between data input and weights, this result suggests that the logits of both ID and OOD data are centered around zero in the logit space prior to model training, as shown in fig. 1b.

**Assumption 1** (Distributional Separation). *Let input data $x \sim \mathcal{P}_x$ and model initial weights $\omega \sim \mathcal{P}_\omega$ be drawn from distinct distributions with*

$$\text{supp}(\mathcal{P}_x) \cap \text{supp}(\mathcal{P}_\omega) = \emptyset \tag{1}$$

**Proposition 1** (Covariance Bound). *Under Assumption 1, the covariance satisfies:*

$$|\text{Cov}(x, \omega)| \leq \epsilon$$

*for small $\epsilon > 0$.*

*Proof.* The disjoint support in eq. (1) almost surely makes their covariance small. □

**Corollary 1.** *For zero-centered initialization ($\mathbb{E}[\omega] = 0$ as in (Glorot & Bengio, 2010)):*

$$|\text{Cov}(x, \omega)| \leq \epsilon$$

$$\left| \mathbb{E}[\langle x, \omega \rangle] - \mathbb{E}[x] \underbrace{\mathbb{E}[\omega]}_{\mathbb{E}[\omega]=0} \right| \leq \epsilon$$

$$|\mathbb{E}[\langle x, \omega \rangle]| \leq \epsilon$$

*Thus, the expected logit magnitude for an initialized DL classifier is $\mathcal{O}(\epsilon)$ small.*

**Allocation of ID logits post-train:** Assuming that DL classifiers operate as spatial-invariance pattern-matching algorithms operating over the distribution of class-specific discriminative features, these models produce high positive logits when the data features exhibit strong alignment with the feature representations parameterized during training. To do so, DL classifiers rely on convolutions and matrix multiplications, which are composed of dot product operations between the model weights and the input data. Training a DL classifier involves utilizing the cross-entropy loss, $\left(i.e., \mathrm{H}(\mathcal{Y}, \hat{\mathcal{Y}}) = -\sum_k \mathcal{Y}(k) \log(\hat{\mathcal{Y}}(k))\right)$, to encourage the prediction $(\hat{\mathcal{Y}})$ to closely align with the ground truth $(\mathcal{Y})$. When employing one-hot encoding for both $\hat{\mathcal{Y}}$ and $\mathcal{Y}$, the training objective simplifies to:

$$\mathrm{H}(\mathcal{Y}, \hat{\mathcal{Y}}) = \underbrace{-\mathcal{Y}(j) \log(\hat{\mathcal{Y}}(j))}_{\mathcal{Y}(j)=1} - \sum_{i, i \neq j} \underbrace{\mathcal{Y}(i) \log(\hat{\mathcal{Y}}(i))}_{\mathcal{Y}(i)=0} = -\log(\hat{\mathcal{Y}}(j)).$$

Eventually, the minimization of the cross-entropy loss $\left(i.e., \min[\mathrm{H}(\mathcal{Y}, \hat{\mathcal{Y}})]\right)$ equivalues to the maximum likelihood estimation (MLE) $\left(i.e., \min[-\log(\hat{\mathcal{Y}}(j))]\right)$.

As training progresses, the softmax layer aims to generate a response close to one for the cell (index $j$) corresponding to the correct class $\left(i.e., \hat{\mathcal{Y}}(j) \to 1\right)$. Additionally, owing to the inherent property that the softmax output is confined within a simplex $\left(i.e., \hat{\mathcal{Y}}(j) + \sum_{i, i \neq j} \hat{\mathcal{Y}}(i) = 1\right)$, the remaining cells (indexed $i$) are pushed towards values close to zero $\left(i.e., \hat{\mathcal{Y}}(i)_{i \neq j} \to 0\right)$.

Hence, optimization can be conceptualized as the maximization of the softmax cell corresponding to the correct class and the simultaneous minimization of cells associated with incorrect classes.

This pattern of maximization-minimization is also observed in other classification losses (i.e., Support Vector Machine (Tang, 2015) and Kullback-Leibler divergence (Cui et al., 2024)), which are commonly employed in training DL classification models. This maximization-minimization optimization extends from softmax cells directly to the respective logit cells, as softmax maintains the order of logits. In particular, the logit cell linked to the correct class tries to attain large positive values (see fig. 1c).

However, when suppressing the negative values in an activation layer, the minimization process results in logit values near zero rather than approaching negative values of high magnitudes (see Proposition 2 in Appendix A). Therefore, ID data are projected toward the positive regions of the logit space (see Proposition 2 in Appendix A). Given that the logits reach their high positive value for the correct logit cell indicated by the one-hot encoding and approach zero for all other categories, it is evident that the logits for ID samples cluster by class along orthogonal axes within the logit space (see Proposition 2 in Appendix A).

This class-wise orthogonality has been previously observed in pre-logit (i.e., $\mathcal{E}$) embeddings (Kothapalli, 2023; Papyan et al., 2020). However, since the final layer performs an affine transformation defined by $\mathbb{L} = \mathcal{E}\mathcal{W}$, sustaining high classification accuracy requires separation in $\mathcal{E}$ to translate into $\mathbb{L}$. By applying induction on this affine property, one can show that class-wise orthogonality is not limited to the embeddings but emerges consistently in the logit space as well.

**Allocation of OOD logits post-train:** During training, the objective is to align the model's weights with the class-specific discriminative features of the ID data. Given the features of OOD and ID data derived from different distributions, they inherently exhibit a certain level of distributional shift in the feature space. This shift is also reflected in the distribution of the OOD features in relation to the model's parameters since the latter are trained to align with the ID discriminative features. This shift implies that the alignment between OOD discriminative features and parameters will likely remain minimal, even post-train. As a result, their expected dot product tends to yield smaller magnitudes. Therefore, OOD data tend to remain centered within the logit space even after training (see figs. 1b and 1c). While some of the near-OOD samples might fall within the ID region, the far-OOD samples remain separated (see fig. 1c).

# 4 RESULTS

In these experiments, we empirically observe that the OOD logits remain near the center of the logit space both before and after training. In contrast, ID logits consistently gravitate towards clusters around class-specific areas in the positive regions of the logit space. Furthermore, we show that these ID clusters align with the orthogonal axis that spans the logit space embeddings. While the majority of the experiments focus on far-OOD data, we additionally evaluate on synthesized near-OOD cases to expose their expected phenomena in the logit space.

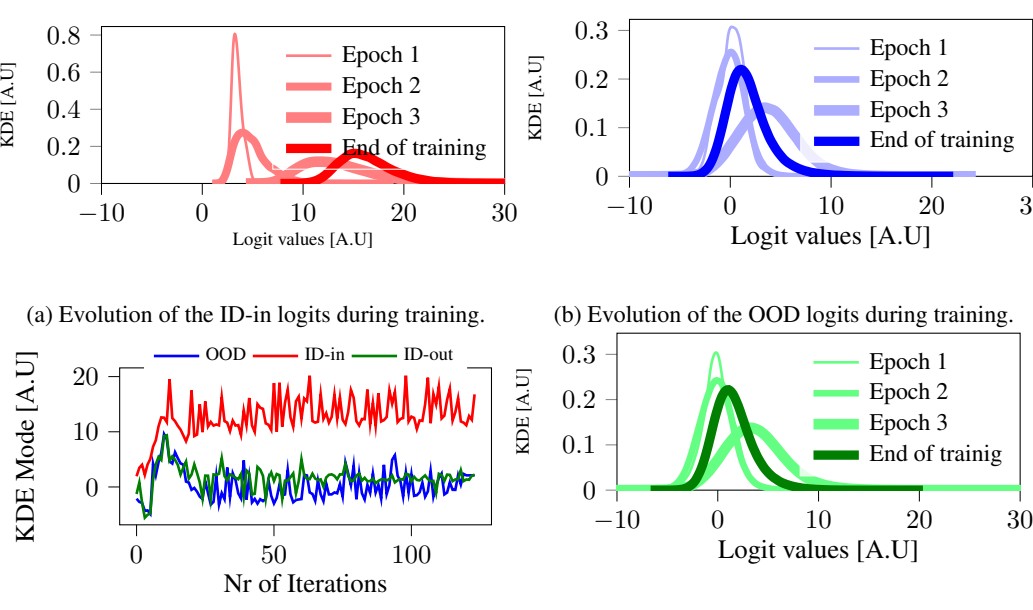

(a) Evolution of the ID-in logits during training.

(b) Evolution of the OOD logits during training.

(c) OOD and ID mode (i.e., density peak).

(d) Evolution of the ID-out logits during training.

Figure 2: Figure 2a presents the density plot across various epochs for the aggregation of ID-in across all logits, while fig. 2b displays the density plot across different epochs for the aggregation of OOD across logits. Similarly, fig. 2d shows the density plot over different epochs for the aggregation of ID-out across all logits. Since the KDE plots are limited to the first three and the final epochs, we included fig. 2c to provide a comprehensive view of the entire trajectory, featuring the peak (i.e, mode) of the density plot for every epoch.

**OOD vs ID during training:** In fig. 2, we empirically illustrate the distribution of ID and OOD logits before and after training. Additionally, we present the evolution of these distributions throughout the training process. To do so, we employed Resnet-9 (He et al., 2016) with CIFAR-100 (Krizhevsky et al., a) as the ID data and CIFAR-10 (Krizhevsky et al., b) as the OOD data. *Additionally, for correctly classified ID data, we define: (1) 'ID-in' as the logit value of the true class (which must be both the maximum value across all logits and aligned with its one-hot encoded label), and (2) ID-out: as all other logit values in the output vector.*

We represent the empirical distributions of the logit outputs for both ID and OOD samples via kernel density estimation (KDE) (Bishop, 2006). At the beginning of training, one can notice that the densities for both OOD and ID logits are concentrated near zero (see figs. 2a to 2d). While OOD and ID-out logits maintain their central tendency around zero (see fig. 2c) the ID-in logits exhibit a shift towards higher positive values (see fig. 2a). Analyzing the peak (i.e., mode) of each KDE density plot (i.e., ID-in, ID-out, and OOD in fig. 2c), it is evident that ID-in trends towards positive values over time as anticipated by Proposition 2 . Furthermore, the ID-out and OOD logits remain centrally positioned, aligning with our analytical predictions. In addition to the density plots in figs. 2a, 2b and 2d, which illustrate the aggregation of ID-in, ID-out from training data along with OOD across all logit cells, see Appendix D for a detailed visualization of density plots on individual logit cells for a more in-depth analysis.

**Objective:** To empirically validate the persistence of this configuration, we conduct a series of comprehensive experiments across diverse settings, including different architectures such as DenseNet,

ResNet, and Vision Transformers, along with the impact of various activation functions and dropout rates. Furthermore, to rigorously analyze the distributional properties of ID and OOD logits, we employ KDE to visualize their densities. This unified visualization framework enables a direct comparison of ID-in (the maximum logit values for correctly classified ID samples), ID-out (the remaining logit values for correctly classified ID samples), and OOD. By overlaying their KDE plots, we assess the shift between ID-in and OOD, the separation between ID-in and ID-out, and the degree of overlap between OOD and ID-out distributions.

**Effect of batch-normalization:** Batch normalization (BN) (Ioffe & Szegedy, 2015) is widely used in DL to stabilize and accelerate training by centering and scaling activations. However, its impact on the separation between ID and OOD logits remains unclear. We examine this in a controlled setting using the same experiment as in Figure 1, while comparing two identical networks, with and without BN applied after every layer (see Appendix B for model details). When BN is applied, the logits of ID samples are pulled closer to the center, whereas the logits

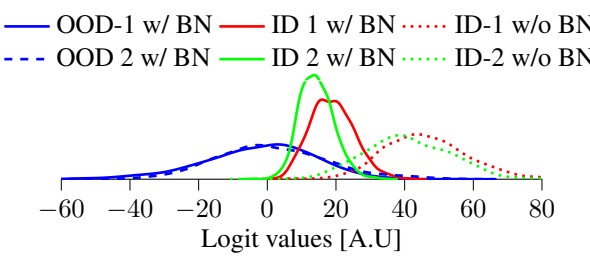

Figure 3: BN on an MLP binary classifier pulls ID logits toward zero, while OOD logits stay centered as in the baseline (w/o BN).

of OOD samples remain clustered around the center, just as they were without BN (see fig. 3). The key driver is the centering step in BN (i.e., $\hat{x}_i = \frac{x_i - \mu_{\mathcal{B}}}{\sqrt{\sigma_{\mathcal{B}}^2 + \epsilon}}$) which subtracts the mean $\mu_{\mathcal{B}}$. For ID activations, this operation merely recenters values that are already positive, so most of them stay above zero. In contrast, OOD activations can fall well below $\mu_{\mathcal{B}}$; after normalization they often become negative and are subsequently clipped to zero by the activation function.

Additional experiments with different dropout rates and different activation functions are included in Appendices E and F.

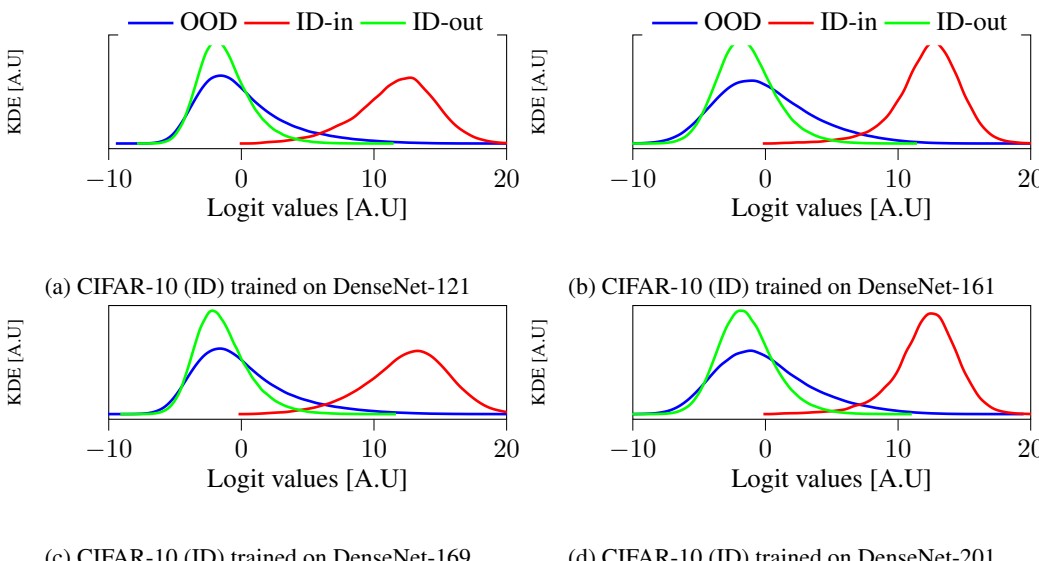

(a) CIFAR-10 (ID) trained on DenseNet-121          (b) CIFAR-10 (ID) trained on DenseNet-161

(c) CIFAR-10 (ID) trained on DenseNet-169          (d) CIFAR-10 (ID) trained on DenseNet-201

Figure 4: An analysis of the density over aggregated logits across distinct DenseNet architectures trained on the CIFAR-10 dataset as the ID data, while the OOD includes $\{\mathcal{D}\} \setminus$ CIFAR-10. For a more detailed comparison, check figs. 34 to 41 in Appendix G.

**Experiments on different CNN classifiers:** The analysis of ID and OOD logits has been expanded across various DL classifier models. Our study examines various iterations of DenseNet (Huang et al., 2017), specifically versions 121, 161, 169, and 201, as well as ResNet (He et al., 2016), encompassing versions 18, 34, 50, 101, and 152. Furthermore, the utilized experimental dataset

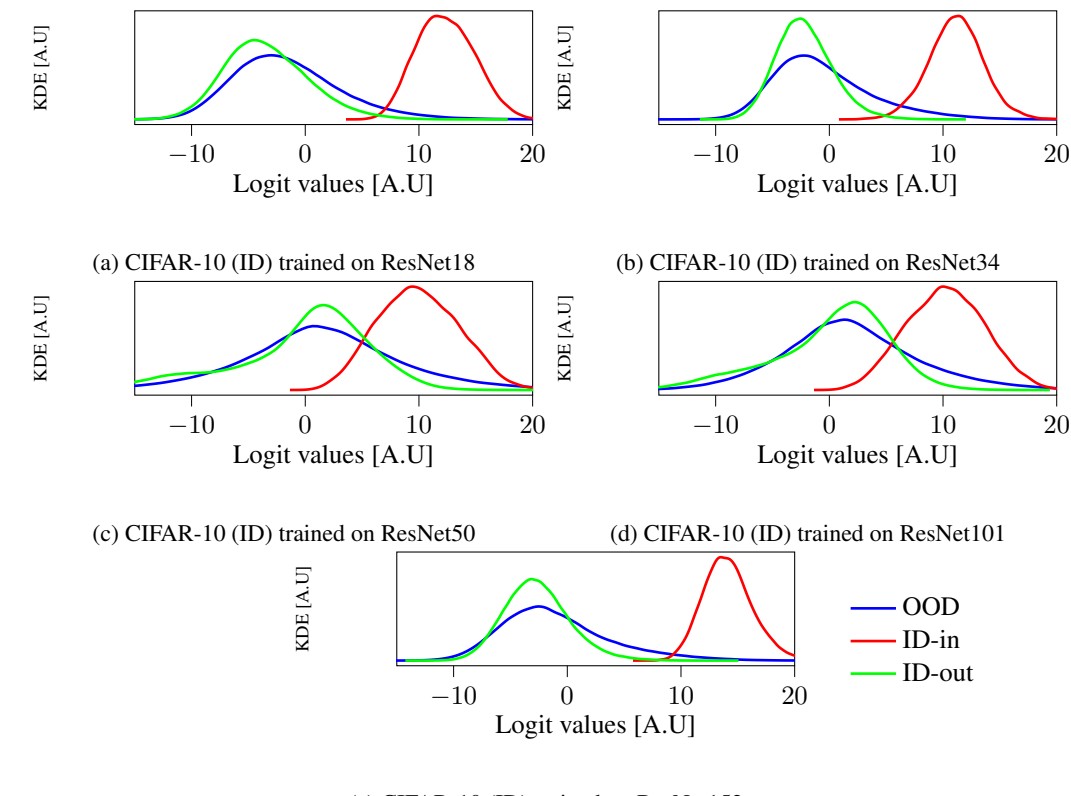

(a) CIFAR-10 (ID) trained on ResNet18

(b) CIFAR-10 (ID) trained on ResNet34

(c) CIFAR-10 (ID) trained on ResNet50

(d) CIFAR-10 (ID) trained on ResNet101

(e) CIFAR-10 (ID) trained on ResNet152

Figure 5: An analysis of the density over aggregated logits across distinct ResNet architectures trained on the CIFAR-10 dataset as the ID data, while the OOD includes $\{\mathcal{D}\} \setminus$ CIFAR-10. For a more detailed comparison check figs. 42 to 51.

comprises $\{\mathcal{D}\} = \{$SVHN, CIFAR-100, CIFAR-10, Tiny ImageNet (Deng et al., 2009a), iSUN (Xu et al., 2015), LSUN (Yu et al., 2016)$\}$. Densenet and ResNet models are trained using SGD with a cyclical learning rate starting at $lr = 10^{-3}$ with a cosine annealing operation with a periodicity of 200. Furthermore, the momentum is 0.9 while the weight decay $5 \cdot 10^{-4}$. A batch size of 256 is applied for both test and train data, while the number of epochs is 200. ReLU is utilized as an activation function for every layer. No regularization is applied to the training process, while the training data are augmented with random flipping and cropping. Each version of Densenet and ResNet undergoes separate training on CIFAR-10 and SVHN as ID datasets. When CIFAR-10 is utilized as ID, the remaining datasets are employed as OOD data, specifically $\{\mathcal{D}\}$ without CIFAR-10 (i.e., $\{\mathcal{D}\} \setminus$ CIFAR-10) is utilized as OOD. Similarly, when SVHN is utilized as ID, the remaining datasets are employed as OOD data, specifically $\{\mathcal{D}\}$ without SVHN (i.e., $\{\mathcal{D}\} \setminus$ SVHN).

Observations indicate that the ID-in logits consistently tend toward higher positive values across various versions of DenseNet (see fig. 4 and fig. 32 in Appendix G) and ResNet (see fig. 5 and fig. 33 in Appendix G). Contrarily, ID-out and OOD logits tend to be concentrated around zero. Interestingly, the spread and the degree of overlap between ID and OOD logits—remain consistent across different model architectures, including both ResNet and DenseNet variants. This suggests that these properties are largely architecture-agnostic, reinforcing the generalizability of our findings.

**Experiments on different vision transformers:** Contrary to traditional convolutional neural networks (CNN) (e.g., DenseNet, ResNet), which process image patches exclusively on a spatial level, vision transformers (ViT) incorporate an additional component of interleaved processing among patches through the attention mechanism (Dosovitskiy et al., 2021). To examine the effects of this interleaved processing on the arrangement of OOD and ID logits, we carried out experiments with various ViT configurations, including the base (ViT-B) and large (ViT-L) models, each with two different patch sizes: 16x16 and 32x32 pixels. Furthermore, the utlized experimental dataset comprises

$\{\mathcal{D}\} = \{$ SVHN, CIFAR-100, CIFAR-10, Tiny ImageNet, iSUN, LSUN$\}$. Each model undergoes separate training on CIFAR-10 and SVHN as ID datasets. The remaining datasets are employed as OOD data, specifically $\{\mathcal{D}\} \setminus$ CIFAR-10 and $\{\mathcal{D}\} \setminus$ SVHN.

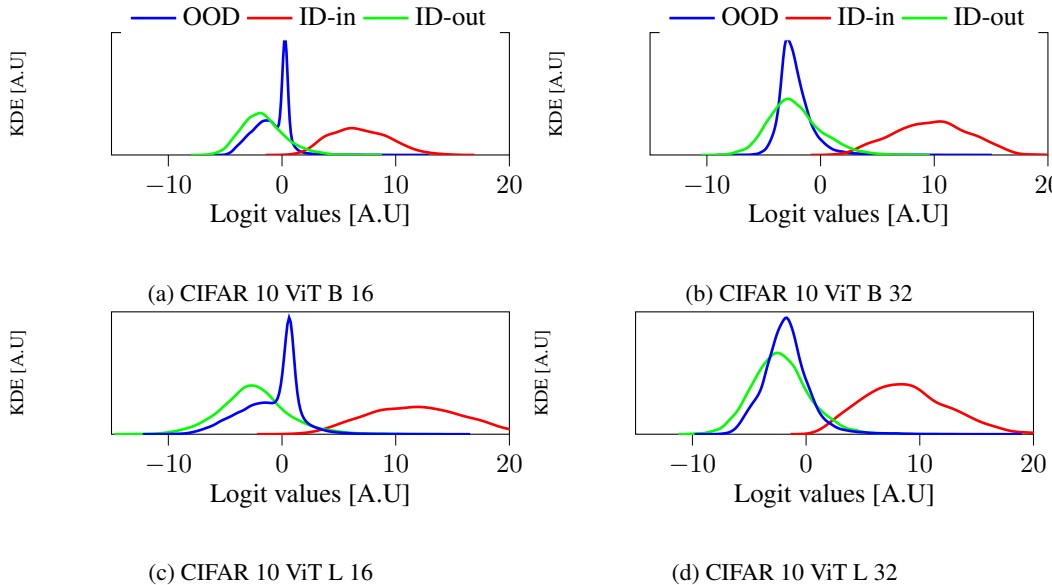

(a) CIFAR 10 ViT B 16

(b) CIFAR 10 ViT B 32

(c) CIFAR 10 ViT L 16

(d) CIFAR 10 ViT L 32

Figure 6: An analysis of the density over aggregated logits across distinct ViT architectures trained on the CIFAR-10 dataset as the ID data, while the OOD includes $\{\mathcal{D}\} \setminus$ CIFAR-10. For a more detailed comparison check figs. 52 to 60 in Appendix H

In fig. 6, one can notice that for all versions of the ViT, ID-in logits converge towards higher positive values as expected. Contrarily, the logits for both the ID-out and OOD samples predominantly cluster around the center of the logit space. The persistence of the anticipated logit configurations for both OOD and ID data empirically observes that, similarly to CNNs, ViTs effectively parameterize the discriminative feature distribution of ID data. While CNNs exclusively leverage localized hierarchical features, ViTs augment these local patterns with global contextual information through self-attention mechanisms (Vaswani et al., 2017). Since self-attention operates via patch-wise dot product interactions, it preserves the intrinsic feature structure of the ID data, avoiding spurious feature generation. Furthermore, varying patch sizes (i.e., 16x16 to 32x32) in ViTs exhibit negligible impact on the resulting logit distributions, suggesting that even the small patch size is sufficient for the global context encoding. This invariance underscores that the parameterization of discriminative features is largely unaffected by patch-wise tokenization, reinforcing the stability of ViTs in modeling ID data distributions. Similarly, scaling the ViT from the base configuration (12 layers, 768 hidden dimensions, 12 attention heads) to the larger variant (24 layers, 1024 hidden dimensions, 16 heads) preserves the overall logit distribution structure. Despite the significantly expanded parameter space, the larger model does not exhibit a substantial improvement in the separability between ID and OOD samples, suggesting that mere architectural scaling alone is insufficient to enhance OOD detection performance.

**Near-OOD Detection Experiments:** While prior work has focused on far-OOD detection (datasets disjoint from ID data), we investigate the more challenging near-OOD regime. We first synthesize very near-OOD samples by interpolating between features of distinct ID classes via:

$$\mathbb{X}_{\text{near-OOD}}^{\text{mixed}} = t \cdot \mathbb{X}_{\text{ID}}^{c_1} + (1 - t) \cdot \mathbb{X}_{\text{ID}}^{c_2}, \tag{2}$$

such that $t \in [0, 1]$, and, $c_1 \neq c_2$, where $t$ controls the mixing strength (Wang et al., 2022).

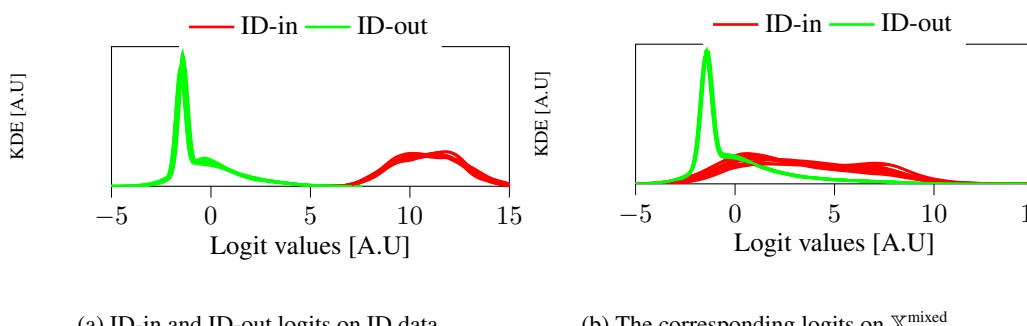

(a) ID-in and ID-out logits on ID data.     (b) The corresponding logits on $\mathbb{X}_{\text{near-OOD}}^{\text{mixed}}$ .

Figure 7: Figure 7a indicates ID-in and ID-out logits for the CIFAR-10 as ID test data when using Resnet-34. Figure 7b indicates the shift of these logits towards the center when $\mathbb{X}_{\text{near-OOD}}^{\text{mixed}}$ data using the mixing operation in eq. (2).

Additionally, we perform linear interpolation with noise,

$$\mathbb{X}_{\text{near-OOD}}^{\text{noise}} = t \cdot \mathbb{X}_{\text{ID}} + (1-t) \cdot \Omega, \quad (3)$$

such that $t \in [0,1]$, and, $\Omega \sim N(0,1)$, to corrupt the images to generate synthetic outliers.

Both of these linear mixing operations (see eqs. (2) and (3)) dilute the discriminatory characteristics of the class (see eq. (2)), causing the resulting samples to exhibit OOD behavior. As shown in figs. 7b and 8, near-OOD samples are closer to the origin of the logit space, unlike ID samples, which form separable class clusters (see fig. 7a). This aligns with our expectation, as class-specific features become diluted during interpolation, their dot product with the model weights decreases, leading to correspondingly lower logit values (red KDE density plots in figs. 7b and 8).

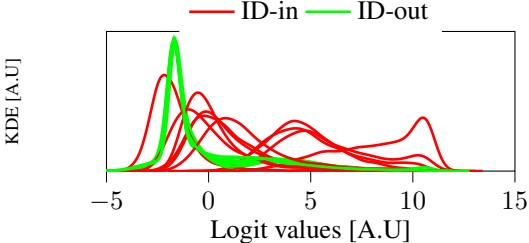

Figure 8: Logits for $\mathbb{X}_{\text{near-OOD}}^{\text{noise}}$ (see eq. (3)); adding noise shifts samples toward the origin in logit space relative to ID.

**ImageNet-1K Experiments:** To confirm that our observations scale to large-scale settings, we repeated our logit-space analysis on ImageNet-1K (Deng et al., 2009b). Using several pretrained models (see Appendix C), we extracted the distributions of ID-in logits (the maximum logits for correctly classified samples), ID-out logits (all remaining logits), and OOD logits—here drawn from the Places (Zhou et al., 2018) and Textures dataset (Cimpoi et al., 2014).

Because ImageNet-1K has 1000 classes, we aggregated all ID-in values into one distribution, all ID-out values into a second, and all OOD logits into a third (fig. 9).

In every case, ID-in logits remain strongly positive, whereas OOD logits cluster near zero. This result, consistent across dataset scale and model depth, demonstrates the robustness of our logit-based separation. As shown in Appendix C, we evaluate a broad suite of architectures—additional ResNet variants, DenseNet, ResNeXt (Xie et al., 2017),

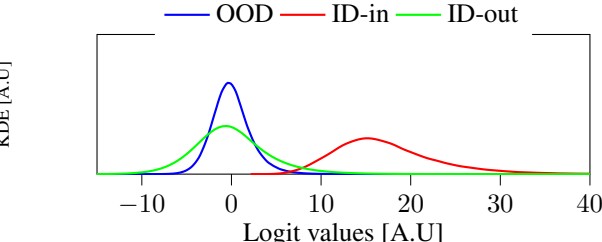

Figure 9: KDE of the "ID-in" logits and "ID-out" logits across all 1000 ImageNet-1K categories from a pretrained ResNet-18, overlaid with the density of OOD samples.

Wide-ResNet (Zagoruyko & Komodakis, 2016), MobileNet (Howard et al., 2017), and ViT—and find that the observed behavior is stable across both convolutional models and vision transformers.

## 5 CONCLUSION

Although current research on OOD detection focuses on developing new methods that naturally give higher scores to ID data and, by default, lower scores to OOD samples, this study concentrates on analyzing the differences between OOD and ID logit distributions.

Specifically, we empirically observed the anticipated configuration of OODs and IDs logits, i.e., that ID logits are clustered by class towards the positive region of the logit space, aligning with the orthogonal axis that spans this space. Additionally, OOD logits remain consistently shifted from ID logits, drawn around the center of the logit space.

This behavior of OOD and ID logits is consistent across various architectures (i.e., CNN and ViT) and activation functions tested on a set of large and diverse OOD data.

As a future direction, the observed patterns within OOD, ID-in, and ID-out logits indicate the potential for a novel approach that leverages ID-out logits as proxies for OOD instances. This approach will facilitate the development of a binary classifier neural network designed to differentiate between OOD and ID samples, employing ID-out logits as representative proxies for OOD instances. Consequently, this method addresses OOD detection as a straightforward classification challenge, thus mitigating the need for threshold-based discrimination methods.

An additional crucial application of the observed logit configuration is the detection of ID data shifts. Since ID values are typically oriented towards positive values along the corresponding axis, this characteristic can be utilized to develop a more accurate and scalable approximation of the Wasserstein distance. Consequently, this enables a more sensitive metric to detect shifts toward the center of the logit space in the ID test data.

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
