# OpenReview forum: "A Geometric Analysis of Logit Embeddings for Out-of-Distribution Detection"
_ICLR.cc/2026/Conference — Submitted to ICLR 2026_

### Official Review · Reviewer_kGiW · 2025-10-20

**Soundness:** 1
**Presentation:** 2
**Contribution:** 1
**Rating:** 2
**Confidence:** 5

**Summary:**

This work proposes an analysis into the logit embeddings between in-distribution (ID) data and out-of-distribution (OOD) data. The key insight is that OOD data is drawn to the center of the logit space, and ID data are repelled from the center, dispersing outward into distinct and class-wise clusters. A lot of KDE plots are provided in this paper, and experiments cover a wide range of aspects.

**Strengths:**

1.	The proposed insight about the spatial distribution between ID and OOD data in the logit space is somehow beneficial.

2.	There are extensive intuitive KDE plots.

**Weaknesses:**

1.	Insufficient workload. The contribution of this work is merely the aforementioned spatial separability between ID and OOD data in the logit space. The workload is clearly insufficient. How can the separability help OOD detection by inspiring new effective detection scores? The authors are suggested to enrich this paper by proposing a relevant detection method with associate detection results instead of just showcasing a phenomenon.

2.	All the references are not in a correct form where the outermost parentheses are missing. Have the authors read the paper themselves? Do the authors feel uncomfortable when seeing so many in-text references without parentheses?

3.	From line 168 to line 174, the ID logits distribution after training is explained, which is an important conclusion in this work. However, there are too few contents and the key Proposition 2 is even left to the appendix. In contrast, there are so many contents from Line 138 to Line 167 talking about something widely-acknowledged. This part should be deeply revised.

4.	All the results are presented in KDE plots, which is not convincing. At least some numerical results should be presented to support the claim.

5.	The claimed spatial distribution of ID and OOD data in the logit space is also not strongly demonstrated in the submission. For example, the authors claim that ID data logits distribute across the orthogonal axes that span the logit space. To validate this, some PCA-based analysis should be provided, such as the reconstruction errors on ID and OOD data in the logit space. However, throughout the paper, there are only KDE plots on logit values and no any geometric empirical investigations. Such empirical results are insufficient and cannot provide a sound support on the claim.

**Questions:**

My questions correspond to the weaknesses above.

---

> ### Author Response · Authors · 2025-11-17
> **Parentheses on the citations**
>
> >**Comment:** All the references are not in a correct form where the outermost parentheses are missing. Have the authors read the paper themselves? Do the authors feel uncomfortable when seeing so many in-text references without parentheses?
>
> We thank the reviewer very much for this. We apologize for the oversight. In the revised manuscript, we have corrected all in-text citations to include the proper outermost parentheses and ensured consistency with the ICLR style guidelines. We appreciate the reviewer’s careful reading and have addressed the formatting throughout the paper.

---

> ### Author Response · Authors · 2025-11-17
>
> > **Comment:** From line 168 to line 174, the ID logits distribution after training is explained, which is an important conclusion in this work. However, there are too few contents and the key Proposition 2 is even left to the appendix.
>
> Thank you for the thoughtful feedback. Our paper is intentionally organized to showcase the breadth of empirical evidence, given that the contribution is a large-scale study across multiple datasets and architectures. To maintain clarity under the page constraints, we placed the full theoretical development—including Proposition 2 and its proof—in Appendices A–B, while keeping its statement, intuition, and implications summarized in the main text (Lines 168–174). This allows readers to access the formal details in a single, coherent location without interrupting the empirical narrative.
>
> > **Comment:** In contrast, there are so many contents from Line 138 to Line 167 talking about something widely-acknowledged. This part should be deeply revised.
>
> Regarding Lines 138–167, although much of this material is widely known, we include it to keep the paper self-contained for the ICLR audience and to precisely align notation and assumptions with our setting. We believe the main body emphasizes the empirical findings and their practical significance, and the appendices provide the complete theoretical underpinnings and additional experiments for interested readers.
>
> We hoped to clarify your feedback and looking forward to additional comments.

---

> ### Author Response · Authors · 2025-11-17
> **Comment regarding numerical results**
>
> >**Comment:** All the results are presented in KDE plots, which is not convincing. At least some numerical results should be presented to support the claim.
>
> We appreciate the reviewer’s feedback.
> Given the breadth of experiments (multiple models and datasets), we used KDE plots to clearly illustrate consistent ID–OOD patterns. To complement these visuals, we now include numerical results in Tables 3 and 4 (Appendix H): median values for the set of ID and OOD across several ResNet and DenseNet variants. These show substantially higher positive medians for ID, while OOD medians remain near the center of the logit space.
> We hope this addresses the concern.

---

> ### Author Response · Authors · 2025-11-17
> **Response to the concern on contribution and geometric evidence.**
>
> >**Comment:** The claimed spatial distribution of ID and OOD data in the logit space is also not strongly demonstrated in the submission. For example, the authors claim that ID data logits distribute across the orthogonal axes that span the logit space. To validate this, some PCA-based analysis should be provided, such as the reconstruction errors on ID and OOD data in the logit space. However, throughout the paper, there are only KDE plots on logit values and no any geometric empirical investigations. Such empirical results are insufficient and cannot provide a sound support on the claim.
>
> We appreciate the reviewer’s request for stronger geometric validation. We considered PCA-based analyses but found them ill-suited for our specific claim for two reasons: (i) PCA requires mean-centering, which collapses class-wise ID structure toward the origin and obscures the axis-aligned separation we study; and (ii) PCA captures directions of maximum variance rather than alignment with class-separable coordinates, making it a weak proxy for assessing the claimed orthogonality.
>
>
> **Our approach establishes the geometry in two steps:**
>
>
>
>
> **From probabilities to class-wise orthogonality.**
> In Appendix B (Proposition 2), we start from the softmax representation, where predictions lie on the probability simplex $\sum_i p_i = 1$. Under the standard maximum-probability decision rule, ID samples concentrate near the vertex corresponding to the true class (high $p_y$, others near zero). This yields class-wise clustering aligned with coordinate axes in probability space.
>
>
> **Transfer to logit space.**
> We relate this structure to logits via the monotone mapping between logits and softmax probabilities. Under typical training dynamics, ID-in logits for the true class move to large positive values while non-true-class logits remain near (not strongly negative) values; OOD logits concentrate near the center. This preserves class-wise axis alignment in logit space (Appendix B, Proposition 2).
>
>
>
>
> **Why KDE instead of PCA.**
> To make the geometry intuitive, we used KDE plots, which consistently show OOD densities near the origin and ID densities shifted toward large positive values along class-specific coordinates. These plots communicate the axis-aligned concentration more directly than PCA would after global mean subtraction.
> We hoped to address your concern and we look forward to additional questions regarding this matter.
>
>
> Thank you for your time.

---

> ### Author Response · Authors · 2025-11-18
> **Comment on Insufficient workload.**
>
> >**Comment:** Insufficient workload. The contribution of this work is merely the aforementioned spatial separability between ID and OOD data in the logit space. The workload is clearly insufficient. How can the separability help OOD detection by inspiring new effective detection scores?
>
> Our goal is to surface and rigorously validate a persistent, architecture- and dataset-agnostic trend in logit space: a clear spatial separability between ID and OOD samples. To our knowledge, this phenomenon has been largely overlooked as a design principle by existing OOD methods. We document it systematically across standard ID datasets (SVHN, CIFAR, ImageNet and grayscale variants such as MNIST/Fashion-MNIST), a broad suite of OOD sources (iSUN, LSUN, Places, Texture), and diverse model families (ResNet, Wide-ResNet, DenseNet, ResNeXt, MobileNet, ViT). Moreover, we investigated separately Batch-Normalization and Dropout.
> The consistency we observe across these settings indicates that the effect is not an artifact of a particular dataset or architecture, but a robust property of modern classifiers.
>
> >**Comment:** The authors are suggested to enrich this paper by proposing a relevant detection method with associate detection results instead of just showcasing a phenomenon.
>
> Documenting the cross-model/data consistency of ID–OOD logit embeddings and introducing a new detection method are both substantial undertakings. To maintain clarity and meet space constraints, this paper concentrates on the empirical finding, which we view as the methodological foundation. We hope this clarifies the scope and value of the current contribution.

---

> > ### Comment · Reviewer_kGiW · 2025-11-25
> >
> > Thanks for providing the detailed clarification. I have carefully read your rebuttal. But most of the rebuttal is repeating what I have known from reading your paper. The following are the most important weaknesses from my perspective.
> >
> > All the empirical results in this work are around the naive logit values. The KDE plots are directly visualizing the logit values, and the supplemented results in your rebuttal are still very naive medians from the logit values. I do not agree that such results can help understand the geometric structure as claimed in your work. It is widely acknowledged that ID samples tend to be assigned with logits that contain a peak value in one label, while OOD samples tend to be assigend with logits that contain uniform values across all labels, which naturally results in the difference in KDE plots and median values. In short, such naive analyzes directly on the logit values are not sufficiently in-depth and cannot support the geometric patterns.
> >
> > An associated OOD detection method is necessary, given that the emiprical finding is not very attractive and well supported. Indeed, lots of network structures and datasets are exploited, but still, only the logit values are visualized. This is why the workload is insufficient.

---

> > > ### Author Response · Authors · 2025-11-26
> > >
> > > >**Comment:** I do not agree that such results can help understand the geometric structure as claimed in your work. It is widely acknowledged that ID samples tend to be assigned with logits that contain a peak value in one label, while OOD samples tend to be assigned with logits that contain uniform values across all labels, which naturally results in the difference in KDE plots and median values. In short, such naive analyzes directly on the logit values are not sufficiently in-depth and cannot support the geometric patterns.
> > >
> > >
> > > We thank the reviewer for their careful attention to the data visualization, and we acknowledge that our presentation of the density is a slight modification of standard practice for multi-dimensional data.
> > >
> > > Our methodology employs a slight variant of standard marginalized density plots. To clearly showcase the structural separation, we made two conscious design choices:
> > >
> > > 1. ID-in Marginalization: We marginalize exclusively over the ID-in (correct class) logit to highlight its persistent high positive values, which contrasts sharply with the other groups.
> > >
> > > 2. ID-out Collapse: Given the high empirical similarity of the ID-out (incorrect class) per-class marginal distributions, we collapse them into a single density plot. This grouping avoids redundancy and simplifies the visualization without losing any density information pertinent to the structural finding.
> > >
> > >
> > > We chose this method as the simplest way to visually capture the core structural contrast: the positivity of the ID-in versus the near-zero nature of the collapsed ID-out densities. Isn't this simple visualization sufficient to showcase the orthogonality of the ID class-wise clusters?
> > >
> > >
> > >
> > >
> > >
> > >
> > >
> > >
> > > >**Comment:** An associated OOD detection method is necessary, given that the empirical finding is not very attractive and well supported. Indeed, lots of network structures and datasets are exploited, but still, only the logit values are visualized. This is why the workload is insufficient.
> > >
> > >
> > > We agree that developing improved OOD detection algorithms is very important.
> > >
> > > However, we believe that showcasing this consistent structural trend across models and datasets is of equal foundational importance.
> > >
> > > The vast majority of existing OOD methods are designed without explicit awareness of this specific geometric configuration, treating the separation as a given rather than a mechanism to be understood.
> > >
> > > By making this underlying structure explicit, we provide the community with a robust, model-agnostic concrete design principle that can inform and simplify the creation of truly principled and effective scoring functions and regularizers in future work.

---

### Official Review · Reviewer_JWp1 · 2025-10-22

**Soundness:** 3
**Presentation:** 3
**Contribution:** 2
**Rating:** 6
**Confidence:** 4

**Summary:**

This paper presents a comprehensive empirical study of In-Distribution (ID) and Out-of-Distribution (OOD) logit behavior in deep learning classifiers. The authors demonstrate that ID and OOD data exhibit distinct geometric patterns in the logit space: ID logits form clusters in positive regions aligned along class-specific orthogonal axes, whereas OOD logits remain centered near zero. The study shows that this logit configuration persists consistently across different architectures and datasets. Furthermore, the authors suggest leveraging ID-out logits as proxies for OOD detection, although the paper does not provide any methodology for practical deployment.

**Strengths:**

1.	The study covers a wide range of architectures and configurations, demonstrating the strong generalizability of the findings.
2.	The study integrates theoretical insights with experimental results, providing mutually consistent support for its conclusions.
3.	The geometric interpretation of the logit-space configuration could potentially serve as a foundation for developing simple OOD detectors.

**Weaknesses:**

1.	Although the empirical study is extensive, it primarily reinforces existing intuitions rather than offering a novel methodological or substantive theoretical contribution.
2.	The paper does not demonstrate that the reported findings can be effectively leveraged for designing an OOD detector.
3.	The text references numerous figures but detailed qualitative descriptions are limited

**Questions:**

1.	The before/after training analysis focuses exclusively on correctly classified ID samples, overlooking misclassified ID instances. However, in practice, the boundary between ID and OOD data often becomes ambiguous precisely because of such misclassified ID examples, which may exhibit logit distributions similar to OOD samples. Addressing this limitation would strengthen the empirical validity of the conclusions and provide a more realistic understanding of logit-space separability in practical OOD detection scenarios.

2.	The study covers a wide range of architectures and configurations, but it remains unclear which design choices most influence logit-space separability between ID and OOD samples. Providing guidelines or empirical insights on which architectures or hyperparameters favor clearer separation would significantly improve the paper’s practical utility.


3.	The paper proposes using ID-out logits as proxies for OOD, but it does not provide experimental validation. Presenting preliminary results for a binary classifier and clarifying the design of such a detector would strengthen the contribution.


4.	Although the paper includes a related work section, it does not clearly situate its new findings within the context of prior research. For example, are there existing OOD detectors that already exploit the observations made in this study? Which insights are genuinely novel compared to previous knowledge? Do any of the results contradict prior publications? Furthermore, the Neural Collapse (NeCo) phenomenon, extensively analyzed in several recent works, also describes the emergence of class-wise orthogonal clusters, though in the feature space rather than in the logit space. It would be valuable to discuss whether the observed logit-space configuration could be a manifestation or an extension of Neural Collapse. Such a connection could provide a theoretical grounding for the empirical patterns reported in this study.



This paper provides a thorough empirical analysis of ID and OOD logit behavior across a wide range of architectures and datasets. The experiments are well-executed, and the geometric patterns observed in logit space are consistent and generalizable, offering valuable insights for future OOD research. However, the study is primarily descriptive: it does not introduce a new method, lacks concrete guidance for OOD detector design, and does not fully situate its findings within existing theoretical frameworks such as Neural Collapse. Despite these limitations, the paper’s clarity, empirical rigor, and breadth of analysis make it a meaningful contribution that could inform and inspire follow-up work.

---

> ### Author Response · Authors · 2025-11-20
> **Design principles for better separability of ID from OOD samples**
>
> >**Comment:** The study covers a wide range of architectures and configurations, but it remains unclear which design choices most influence logit-space separability between ID and OOD samples. Providing guidelines or empirical insights on which architectures or hyperparameters favor clearer separation would significantly improve the paper’s practical utility.
>
>
> Thank you for your insightful comments and indeed, design principles for ML that can better separate the ID from OOD samples.
> Our findings suggest that certain configurations, such as Batch-Normalization and a high Dropout rate, may actually impair performance in this regard.
>
> Unfortunately, our current analysis has not conclusively determined which specific model types excel at separating ID from OOD samples.
>
> However, we hypothesize in the paper that models which better parameterize class-specific features, thus improving ID per-class classification accuracy, tend to produce ID embeddings that are more distanced from the center, enhancing their separability from OOD data.

---

> ### Author Response · Authors · 2025-11-20
> **Comment on correctly classified ID and misclassified ID samples**
>
> >**Comment:** The before/after training analysis focuses exclusively on correctly classified ID samples, overlooking misclassified ID instances. However, in practice, the boundary between ID and OOD data often becomes ambiguous precisely because of such misclassified ID examples, which may exhibit logit distributions similar to OOD samples. Addressing this limitation would strengthen the empirical validity of the conclusions and provide a more realistic understanding of logit-space separability in practical OOD detection scenarios.
>
>
> We thank the reviewer for this thoughtful comment.
>
> Notice that a key requirement for achieving separation of ID samples from OOD ones is that the classifier must attain high ID accuracy, hence the model can push ID samples away from the center of the distribution (and thus away from OOD embeddings).
>
> In contrast, misclassified ID samples represent only a small portion of the data.
> While they do not substantially shift the overall structure of the ID per-class clusters (as they tend to be in minority), they do contribute to increased overlap with the OOD embeddings.
>
> We are actively investigating this property in follow-up work, where we design a binary ID–OOD detection model that simultaneously improves ID classification accuracy. In this setup, misclassified ID samples are classified as OOD as sometimes they tend towards the center.

---

> > ### Comment · Reviewer_JWp1 · 2025-11-21
> >
> > While misclassified ID samples may represent a minority of the data, they are precisely the cases that make ID–OOD separation challenging in practical scenarios. Minimizing their importance risks overlooking the portion of the distribution where ID–OOD ambiguity is most critical. Clearer justification for excluding these samples from the main analysis—or additional experiments explicitly addressing them—would strengthen the empirical conclusions.

---

> ### Author Response · Authors · 2025-11-24
> **Connection to related work**
>
> >**Comment:** Although the paper includes a related work section, it does not clearly situate its new findings within the context of prior research.For example, are there existing OOD detectors that already exploit the observations made in this study?
>
> While no current OOD detection method explicitly exploits this observed structural pattern, existing works implicitly rely on the belief that ID and OOD embeddings reside in distinct regions. Because this structural separation is not explicitly defined or utilized, most prevailing OOD methods tend to be overly complicated and difficult to formally validate.
> Which insights are genuinely novel compared to previous knowledge?
> The central contribution of this work is the structural separation observed between OOD and ID samples, and the explanation for the emergence of this structure. Although the separation itself is a conventional assumption within OOD detection methods, the specific underlying mechanisms of how and why these samples are separated have not previously been showcased.
>
>
> >**Comment:** Do any of the results contradict prior publications?
>
>
> Not to our knowledge, we were not able to came across any published work that contradicts our results
>
>
> >**Comment:** Furthermore, the Neural Collapse (NeCo) phenomenon, extensively analyzed in several recent works, also describes the emergence of class-wise orthogonal clusters, though in the feature space rather than in the logit space. It would be valuable to discuss whether the observed logit-space configuration could be a manifestation or an extension of Neural Collapse. Such a connection could provide a theoretical grounding for the empirical patterns reported in this study.
>
>
> We thank the reviewer for pointing out this interesting connection to the \textbf{Neural Collapse (NeCo) phenomenon}. The relationship is indeed relevant to our findings. Given that the final Multi-Layer Perceptron (MLP) layer performs an affine transformation, sustaining high accuracy inherently requires class-wise separation in the embedding space ($\mathcal{E}$) and, consequently, in the logit space ($\mathbb{L}$). Specifically, the logits are defined by the affine map $\mathbb{L} = \mathcal{E}\mathcal{W}$. We can prove, using induction based on this affine property, that \textbf{class-wise orthogonality} emerges in both the embedding space $\mathcal{E}$ and the logit space $\mathbb{L}$. We agree that this connection is significant and will include a detailed discussion of its implications in the final version of the paper.

---

> ### Author Response · Authors · 2025-11-24
> **Validations for ID-out logits as proxies for OOD**
>
> >**Comment:** The paper proposes using ID-out logits as proxies for OOD, but it does not provide experimental validation. Presenting preliminary results for a binary classifier and clarifying the design of such a detector would strengthen the contribution.
>
>
> We thank the reviewer for the valuable feedback.
>
> While we agree that achieving superior OOD detection performance is the ultimate goal of this line of research, we deliberately chose to keep the contribution of this paper distinct from proposing a novel OOD detection method.
>
> Our core focus is the theoretical establishment of a robust, model- and dataset-agnostic structural separation between ID and OOD embeddings.
> We emphasize that this explicit structural finding serves as a necessary, concrete design principle for multiple future directions beyond just using ID-out logits as proxies for OOD .
>
> We believe that making this fundamental structure explicit offers a fresh insight vital for improving existing OOD methods and proposing more principled new ones.

---

### Official Review · Reviewer_tTxB · 2025-10-31

**Soundness:** 2
**Presentation:** 1
**Contribution:** 1
**Rating:** 0
**Confidence:** 4

**Summary:**

This paper studies the phenomenon that the magnitude of the logits for OOD samples are typically smaller than that of the ID samples. The mechanism behind this phenomenon is explained by analysing the loss. In the experiments, several networks are analyized by showing the difference in logit distributions of OOD/ID samples.

**Strengths:**

* In the experiments, many networks, including CNNs and vision transformers, are analyzed.
* The writing is easy to follow.

**Weaknesses:**

This phenomenon is not new, and it has been the reason for the design of the very first few OOD algorithms, such as MSP and max logit. Some later works [a] already explicitly designed a regularizaion to encourage the logit-smallness of OOD samples. The ID clustering structure is also used in [b] to desgin better OOD scores. Besides, the paper does not provide new OOD scores based on the gained insights. The experiments also lack quantitative results.


- [a] Training confidence-calibrated classifiers for detecting outof-distribution samples. ICLR 2018
- [b] MOS: Towards scaling out-ofdistribution detection for large semantic space. CVPR 2021

**Questions:**

/

---

> ### Author Response · Authors · 2025-11-24
> **Comment regarding the novelty of the work**
>
> >**Comment:** This phenomenon is not new, and it has been the reason for the design of the very first few OOD algorithms, such as MSP and max logit.
>
> We thank the reviewer for the feedback.
>
> We are familiar with the seminal Max Softmax Probability (MSP) work (Hendrycks & Gimpel, ICLR 2017).
>
> While the notion of separation of OOD and ID embeddings is known underlying these and subsequent OOD detection algorithms, our contribution is novel because it explicitly identifies the structural mechanism—how and why—this separation occurs.
>
> MSP work does not explicitly state how OOD and ID are separate from one another.
>
> Moreover, we are not aware of any work using max logit.
> Moreover, it is very important to notice that because of scale invariance in softmax, max softmax probability does not translate into maximum logit
> $\sigma(z + c1)_i =\frac{e^{z_i + c}}{ \sum_j e^{z_j + c}} = \frac{e^{z_i} e^{c}}{ \sum_j e^{z_j} e^{c}} = \frac{e^{c} e^{z_i}}  {e^{c}\sum_j e^{z_j}} = \frac{e^{z_i}}{\sum_j e^{z_j}}  = \sigma(z)_i $.
>
> We hope to address your concerns and looking forward your feedback.

---

> ### Author Response · Authors · 2025-11-24
> **Comment regarding the existing works**
>
> >**Comment:** Some later works [a] already explicitly designed a regularizaion to encourage the logit-smallness of OOD samples.  [a] Training confidence-calibrated classifiers for detecting out-of-distribution samples. ICLR 2018
>
>
> While having uniform softmax output (low confidence for OOD samples) has been the design principles for many works such as ODIN and including [a] this however does not translate into small OOD logit values.
> The reason is because the softmax output reside on  a simplex $\sum_i p_i=1$ and the scale invariance of the softmax, unform softmax probability does not necessarily translate into minimum logit values.
>
>
> Moreover, fig 1 in [a] does not suggest concentration of OOD near the center of the logit space nor does the work ever mention minimum logit for OOD but rather uniform spread of the OOD embeddings (which is not the case).
>
>
>
>
>
>
>
>
>
>
> >**Comment:** The ID clustering structure is also used in [b] to desgin better OOD scores.  [b] MOS: Towards scaling out-of-distribution detection for large semantic space. CVPR 2021
>
>
> Yes, while ID class-wise cluster is not a novel finding the actuall structure of these clusters in the logit space, namely orthogonal towards the positive values is something that we propose.
>
>
> Moreover, Figure 3 in [b] does not propose any orthogonality of the ID class-wise clusters nor the concentration of OOD near the center of the embedding space.
>
>
>
>
> >**Comment:**  The experiments also lack quantitative results.
>
>
> Given the number of performed experiments we concluded that KDE plots would provide the most straightforward introduction of the observed phenomena. However we did included some quantitative results on Table 3 in appendix H to complement experiments on different versions of ResNet and DenseNet.

---

> ### Author Response · Authors · 2025-11-26
> **Comment on new OOD scoring methods**
>
> >**Comment:** Besides, the paper does not provide new OOD scores based on the gained insights.
>
> While improving OOD detection methods remains important task, we argue that establishing this consistent, universal structural pattern is equally important.
>
> The existing OOD community largely disregards this geometric configuration in their design principles, operating solely on the assumption of separation.
>
> Our contribution explicitly fills this critical gap, providing a robust, foundational characterization that is a necessary prerequisite for developing truly principled and effective OOD scoring and regularization methods.

---

### Official Review · Reviewer_a6xB · 2025-11-04

**Soundness:** 2
**Presentation:** 2
**Contribution:** 1
**Rating:** 2
**Confidence:** 3

**Summary:**

This paper studies learning from out-of-distribution data, and proposes a new perspective of geometric analysis. It is inspired by the limitation of existing OOD detection methods, which usually employs a scoring technique and may neglect the insight within the latent space. In this paper, the authors analyze the logit embedding distributions of ID and OOD data and reveal that the OOD data tends to cluster near the origin.

**Strengths:**

- The studied problem is meaningful.
- The authors conduct extensive experiements to reveal the phenomenum.

**Weaknesses:**

- The contribution of this paper is limited. The main paper consists of many experimental results. However, there are no in-depth analysis regarding why the logit distributions of ID and OOD data exhibits such different phenomenum.
- After revealing the empirical discovery, there are no relevant algorithms proposed to further improve the OOD detection performance. This will also limit the contribution of this paper.
- There are no theoretical analysis provided.
- The experiments are only conducted on some general datasets (e.g., SVHN, CIFAR, ImageNet), which will limit the universality of the proposed method.

**Questions:**

- Is there any existing methods analyzing the OOD detection problem from the latent space?

---

> ### Author Response · Authors · 2025-11-17
> **Scope and value of our contribution**
>
> > **Comment:** After revealing the empirical discovery, there are no relevant algorithms proposed to further improve the OOD detection performance. This will also limit the contribution of this paper.
>
> We thank the reviewer for time.
> We agree that the ultimate goal is better OOD detection performance. However, our contribution in this submission is to establish a robust, model- and dataset-agnostic characterization of how ID and OOD embeddings separate—evidence that to our knowledge, existing methods neither assume nor leverage. We believe making this structure explicit is a necessary that can serve as a concrete design principles for developing principled scoring functions, calibration strategies, and training-time regularizers in future work.
>
> We hope this clarifies the scope and value of our contribution.

---

> ### Author Response · Authors · 2025-11-17
> **Comment regarding theoretical analysis in the Appendix**
>
> >**Comment:** There are no theoretical analysis provided.
>
> We do provide theoretical analysis, split into two parts and detailed in Appendices A and B.
>
> **(1) Why OOD embeddings concentrate near the center of logit space.**
>
>  In Appendix A, under Assumption 1—the data and model weights are drawn from different distributions with disjoint support ($\mathrm{supp}(\mathcal P_x)\cap \mathrm{supp}(\mathcal P_\omega)=\emptyset$)—we show that the expected dot product between inputs and weights is small: $\left|\mathbb{E}[\langle x,\omega\rangle]\right|\le \epsilon$ (Corollary 1). After training, the network’s weights specialize to ID features; OOD features remain distributionally mismatched to these class-specific directions, so the same assumption applies to OOD inputs. Consequently, $\left|\mathbb{E}[\langle x_{\text{OOD}},\omega\rangle]\right|$ is small, yielding logits near zero and embeddings concentrated around the origin.
>
> **(2) Why ID embeddings move away from the center and form classwise structure.**
>
>  At initialization, ID data also satisfy Assumption 1 and behave OOD-like (near the center). During training, the model learns class-specific directions. In Appendix B (Proposition 2), exploiting the central role of dot products and the effect of sign-suppressing activations (e.g., ReLU), we prove that optimization promotes classwise clustering along approximately orthogonal axes. This explains the emergence of separated ID clusters away from the center.
> Together, these results provide a theoretical foundation for the empirical KDE patterns reported in the paper.
> We hoped to address your concern and are looking forward to additional feedback.

---

> ### Author Response · Authors · 2025-11-17
> **In-depth analysis about the distributions of ID and OOD**
>
> >**Comment:** The contribution of this paper is limited. The main paper consists of many experimental results. However, there are no in-depth analysis regarding why the logit distributions of ID and OOD data exhibits such different phenomenum.
>
>
> We thank the reviewer for the opportunity to clarify our stance.
>
> As we wanted to emphesize that in Section 2, in the Method section we do an a general explanation as follows.
>
> In a nutshell,  DL models try to parameterize the distribution of class-specific, invariant, discriminative features from ID training. This yields linearly separable, class-centered clusters in the model’s logit/representation space for ID samples.
>
> By contrast, OOD inputs lack the class-specific features that the model has been trained to amplify. Consequently, their representations are poorly aligned with any class template and tend to fall away from class directions and toward the center of the logit space. Formally, if logits are driven by dot-products between learned class vectors and input features, a distributional mismatch (OOD features vs. ID-trained parameters) reduces the expected alignment, implying lower covariance and hence smaller expected dot-products—which directly manifests as lower-magnitude logits.
>
> This geometric/optimization view explains why ID and OOD logit distributions differ systematically: ID samples align with their class vectors (higher, more peaked logits), whereas OOD samples, lacking aligned features, concentrate near the origin with lower logits.

---

> ### Author Response · Authors · 2025-11-17
> **Existing methods for OOD detection**
>
> > **Comment:** Is there any existing methods analyzing the OOD detection problem from the latent space?
>
> Yes, there is substantial literature analyzing OOD detection in the latent (embedding) space of trained classifiers.
>
> The methods that we explain in Section 2 and Appendix, most methods are effectively ID-first: they model in-distribution behavior (e.g., via feature density, distance, or confidence surrogates) and flag deviations as OOD. This perspective typically leaves the mechanism of ID–OOD separation underexplored, often resulting in complex pipelines and making rigorous, cross-dataset validation difficult.
>
> Our work clarifies this design space by showing that ID and OOD samples occupy consistent and distinguishable regions in logit space. Building on this observation, we shift the focus from exclusively modeling ID behavior to bringing design principles for both ID and OOD data.

---

> ### Author Response · Authors · 2025-11-17
> **Additional experiment on Genome dataset**
>
> >**Comment:** The experiments are only conducted on some general datasets (e.g., SVHN, CIFAR, ImageNet), which will limit the universality of the proposed method.
>
> To further strengthen the case beyond standard natural-image benchmarks, we are adding an additional experiment on a public genomics dataset (Appendix J). The results there replicate the core trend, supporting that our method is both model- and dataset-agnostic.
>
>
> Notice that, our choice of ID datasets (SVHN, CIFAR, ImageNet, and grayscale variants such as MNIST and Fashion-MNIST) followed common practice to ensure comparability with prior work. To stress-test generalization, we paired these with a broad suite of OOD sources—iSUN, LSUN, Places, and Texture—spanning diverse scene/object distributions and low-level statistics. We also evaluated across multiple architectures (ResNet,Wide-ResNet, DenseNet, ResNext, MobileNet  and ViT families), and observed the same, consistent ID–OOD pattern throughout.

---

### Author Response · Authors · 2025-12-04
**General Response to All Reviewers**

We thank all reviewers for their time, careful reading, and constructive feedback. Below, we address the common concerns raised across the reviews.

**1. Novel of OOD Detection Method (Reviewers a6xB, JWp1, kGiW)**

We acknowledge the reviewers' perspective that the ultimate goal of this field is to have better OOD detection methods. However, we firmly believe that **establishing the structural separation** of OOD from ID embeddings is of equal, if not greater, foundational importance.

While existing state-of-the-art (SOTA) methods implicitly assume separation, our comprehensive review suggests that the specific **geometric configuration** we observe (concentration near the origin) has been largely overlooked. By making this structure explicit, we provide a robust **concrete design principle** that can guide the development of more principled scoring functions and regularizers in future work, rather than relying on heuristic improvements.

**2. Additional Experiments beyond Images (Reviewer a6xB)**

To address the concern regarding domain generalization, we have expanded our experimental validation beyond standard image datasets. We have added a section in the **Appendix J** detailing experiments on a **genomics dataset**. These results confirm that the observed structural separation is not an artifact of image data but a fundamental property of the training dynamics.

**3. Novelty of the Structural Observation (Reviewer tTxB)**

We respectfully distinguish between the \textit{assumption} of separation and the \textit{structural mechanism} of separation. While the community has long assumed that OOD and ID samples are separated (as noted by the reviewer), our work explicitly characterizes **how** they are separated in the embedding space.

**4. Connection to Neural Collapse (Reviewer JWp1)**

We appreciate the reviewer highlighting the connection to the Neural Collapse (NeCo) phenomenon. We agree that the class-wise orthogonality we observe in the logit space is intimately linked to similar phenomena in the pre-logit space. We have updated the **Method** section to explicitly discuss this relationship and have included the relevant references to contextualize our findings within the broader NeCo literature.

**5. Theoretical Guarantees (Reviewer a6xB)**

In response to the request for deeper theoretical grounding, we have moved key portions of our analysis from the Appendices into the main **Method** section. Specifically, we now present the core theoretical argument for the initial collapse of both ID and OOD embeddings towards the center, and the subsequent dynamics where training pulls ID embeddings away while OOD embeddings remain centrally located. Supporting proofs and complementary details remain in Appendix A.

**6. Density Visualization / KDE Plots (Reviewer kGiW)**

We agree with the reviewer that standard KDE plots can sometimes obscure high-dimensional nuances. However, given the remarkable consistency of the structural form across diverse models and datasets, we adhered to Occam's Razor, opting for a visualization that maximizes clarity without sacrificing information.

To address the high-dimensionality, we employed a specific modification of standard marginalization:

a) **ID-in Marginalization:** We marginalize exclusively over the correct class logit to highlight its persistent high positive values.

b) **ID-out/OOD Collapse:** Given the high empirical similarity and near-zero nature of the ID-out (incorrect class) and OOD marginals, we collapse them into a single density plot.

This grouping avoids visual clutter and effectively highlights the primary finding: the sharp contrast between the positive ID-in direction and the near-zero concentration of all other signals.


We remain convinced that explicitly characterizing this overlooked structural separation provides a vital missing link for the community, thereby paving the way for more effective OOD detection methods.

---

### Meta-Review · Area_Chair_aB64 · 2025-12-02

**Summary:**

The submission presents an empirical study of logit-space geometry for OOD detection. While the topic is relevant, all reviewers raised substantial concerns regarding **novelty, depth of analysis, and contribution**. **Across most reviewers, the initial scores were low (0–2)**, and even after reading the rebuttal, no reviewer indicated that their major concerns were resolved. The paper remains primarily descriptive, lacks a convincing theoretical or methodological advance, and does not demonstrate how the empirical findings can lead to effective OOD detection methods.

**Reviewer Concerns:**

Across reviewers, several major concerns remain unresolved:

Multiple reviewers noted that the key empirical phenomenon (ID logits having a dominant coordinate and OOD logits being closer to uniform) has been known and used since MSP/ODIN. The rebuttal did not provide convincing evidence of conceptual novelty.

Reviewer JWp1 explicitly requested analysis of misclassified ID samples, which is crucial for realistic OOD scenarios. The rebuttal acknowledged this but provided no empirical examination.

**Reviewer Scores:**

**Reviewer a6xB (score 2)**: The rebuttal did not address their concerns about missing theoretical depth, lack of methodological contribution, and limited generality. They are unlikely to increase the score.

**Reviewer tTxB (score 0)**: This reviewer strongly questioned novelty and contribution, and explicitly reaffirmed these concerns during the discussion. The rebuttal did not provide new evidence that would change their stance. The score would remain 0.

**Reviewer JWp1 (score 6)**: This reviewer was the only borderline-positive opinion. However, their detailed follow-up comment indicates that major concerns regarding misclassified ID samples, methodological impact, and theoretical positioning remain. Even if they might have slightly adjusted phrasing, it is unlikely that they would move the score into a clearly accepting range.

**Reviewer kGiW (score 2)**: This reviewer reiterated in discussion that the rebuttal did not resolve their central objections on workload, geometric justification, and insufficient empirical support. Their score would not increase.

---

### Decision · Program_Chairs · 2026-01-26

Reject